# Repeated semen exposure decreases cervicovaginal SIVmac251 infection in rhesus macaques

Shaheed A. Abdulhaqq[1], Melween Martinez[2], Guobin Kang[3], Idia V. Rodriguez[2], Stephanie M. Nichols[2], David Beaumont[4], Jocelin Joseph[1], Livio Azzoni[1], Xiangfan Yin[1], Megan Wise[1], David Weiner[1], Qin Liu [1], Andrea Foulkes[5], Jan Münch [6], Frank Kirchhoff[6], Christos Coutifaris[7], Georgia D. Tomaras[4], Carlos Sariol[2], Preston A. Marx[8], Qingsheng Li [3], Edmundo N. Kraiselburd[2] & Luis J. Montaner [1]

Semen is the vehicle for virion dissemination in the female reproductive tract (FRT) in male-to-female HIV transmission. Recent data suggests that higher frequency semen exposure is associated with activation of anti-HIV mechanisms in HIV negative sex workers. Here, we use a non-human primate (NHP) model to show that repeated vaginal exposure to semen significantly reduces subsequent infection by repeated low-dose vaginal SIVmac251 challenge. Repeated semen exposures result in lower CCR5 expression in circulating CD4+ T-cells, as well as higher expression of Mx1 (in correlation with IFNε expression) and FoxP3 in the cervicovaginal mucosa, and increased infiltration of CD4+ T-cells. Establishing in vivo evidence of competing effects of semen on transmission impacts our basic understanding of what factors may determine HIV infectivity in humans. Our results clearly indicate that repeated semen exposure can profoundly modulate the FRT microenvironment, paradoxically promoting host resistance against HIV acquisition.

---

[1] The Wistar Institute, Philadelphia, PA, USA. [2] Caribbean Primate Research Center and Animal Resources Center, University of Puerto Rico (UPR), San Juan, United States. [3] School of Biological Sciences University of Nebraska, Lincoln, NE, USA. [4] Duke Human Vaccine Institute and Department of Surgery, Immunology and Molecular Genetics and Microbiology, Duke University, Durham, NC, USA. [5] Mount Holyoke College, South Hadley, MA, USA. [6] Institute of Molecular Virology, Ulm University Medical Center, Ulm, Germany. [7] University of Pennsylvania, Philadelphia, PA, USA. [8] Tulane National Primate Research Center, and Department of Tropical Medicine, School of Public Health and Tropical Medicine, Tulane University, New Orleans, LA, USA. Correspondence and requests for materials should be addressed to L.J.M. (email: montaner@wistar.org)

ntravaginal non-human primate (NHP) SIV challenge studies are the principal pre-clinical model used to model prophylactic interventions in women. To date, these studies typically do not consider the impact of repeated semen exposure on the cervicovaginal environment and its potential effect on viral acquisition. Intravaginal semen exposure induces an acute influx of CD4+ T-cells and other immune cells into the cervical mucosa[1]. Additionally, sustained high-level intravaginal semen exposure (five or more sexual events per week) in sex worker women can lead to cervicovaginal changes such as retention of CD4+ T-cell infiltrates and higher expression of antiviral factors (interferon-epsilon and MX1)[2]. The balance of the effects of semen exposure on HIV-1 or SIV infection remains controversial. In vitro studies reveal a dichotomous role for semen both enhancing[3] and supressing[4] HIV-1 infection of CD4+ T-cells. However, in NHP studies, the presence of semen during acute vaginal exposure to SIV did not significantly affect the rate of infection[5,6]. In contrast, inactivated SIV alone stimulates cervical epithelial cells to recruit myeloid and CD4+ T-cells within 72 h[7], resulting in a more aggressive loss of CD4+ T-cells after intact SIVmac251 infection[8]. It has remained unknown if there are any in vivo effects of repeated semen exposure on the future susceptibility of cervicovaginal HIV infection.

To address the lack of data on the effect of repeated semen exposure before and during the infectious viral challenge and the potential competing effects between semen and viral particle conditioning of the female reproductive tract (FRT), we use a macaque model of repeated semen exposures with or without defective SIV viral particles. With this model, we seek to test whether repeated exposures of the cervicovaginal environment to semen, with or without non-infectious viral particles, could affect tissue susceptibility to infection with replication-competent SIV.

## Results

**Cervicovaginal conditioning with semen/SIVsmB7.** Multiple studies of sero-discordant couples have shown that, despite persistent HIV-1 exposure, male-to-female transmission is a low-frequency event[9,10]. To model repeated semen exposures, we used a single pool of human semen [30% volume] which was administered intravaginally to rhesus macaques twice weekly over 20 weeks. The rate of exposure was chosen to best approximate "average" human sexual activity[11]. In order to include a group with viral exposure without infection, replication-incompetent SIVsmE660 (SIVsmB7[12]) grown in the human-derived CEM cell line was used. Persistent SIVsmB7 and semen exposure was modeled in Group 1 using an inoculum of 500 ng p27 SIVsmB7 with 30% volume human pooled semen. Group 2 regimen only applied pooled human semen [30% volume] in mock CEM supernatant treatment (CEM$_{sup}$); Group 3 regimen excluded semen (replaced with RPMI1640) but maintained SIVsmB7; and Group 4 regimen utilized CEM$_{sup}$ alone (Supplementary Fig. 2 and Supplementary Table 1). We had previously shown that CEM$_{sup}$ has near 100% the identical protein composition as SIVsmB7, with the exception of SIV proteins[8], thus all groups included the presence of human protein content inclusive of CEM$_{sup}$ MHC-I/II. Importantly, challenge studies that followed used SIVmac251 grown in rhesus PBMC devoid of any human MCH-I/II antigens within viral particles.

As detailed in Supplementary Table 1, following the 20 week conditioning period, we performed an escalating dose of intravaginal challenges with SIVmac251 (200–400 TCID$_{50}$) in all 32 animals (8 animals per group) (Fig. 1a and Supplementary Table 1) while maintaining the twice-weekly conditioning days uninterrupted. SIVmac251 challenge was introduced during the first weekly conditioning time-point over a 16-week period (with

a rest week every 3rd week) until detection of plasma SIV RNA (Methods). Therefore, animals conditioned with semen had their challenge dose diluted in the continued presence of 30% volume semen. Animals not conditioned with semen had their challenge dose diluted in RPMI 1640. All animals were maintained on their conditioning regimen throughout the duration of the challenge period until the infection was evidenced. The four groups had similar distribution of age, weight, MHC-I genotype (Table 1) and menstrual status at infection (Supplementary Fig. 1).

**Chronic semen lowers cervicovaginal SIVmac251 infection.** Infection outcomes over the 16 week challenge period were analyzed by comparing all animals conditioned with SIVsmB7 to animals conditioned with semen (16 vs. 16: group 1 + 3 vs. 2 + 4 or 1 + 2 vs 3 + 4, respectively). To confirm that the infection outcomes analyzed were independent of potentially protective intravaginal antiviral responses, anti-SIV responses were analyzed in time-points immediately preceding a positive infection by plasma SIV RNA. As an outcome, one animal in the SIVsmB7 treatment group (9H0) was identified as having developed SIV-specific IgA cross-reactive to both SIVmac and SIVsmm (Supplementary Fig. 3B), and therefore was censored from final analysis. Another animal (BJ39) in the same grouping developed peripheral blood SIV-specific T-cell responses prior to infection (Supplementary Fig. 3A) yet this animal was not censored due to a lack of evidence that CD8+ T cell responses alone can be protective from vaginal challenge.

Using a log-rank test, no difference in infection outcomes was observed between the animals conditioned with SIVsmB7 (11/15 infected) vs. non-SIVsmB7 animals (10/16 infected) (Fig. 1b $p$ = 0.7314). In contrast, semen conditioning conferred resistance to intravaginal SIVmac251 challenge when compared to non-semen conditioned animals (Fig. 1c; log-rank $p$ = 0.0332). Overall, the median infection-free survival time was longer for semen-conditioned animals (week 15) as compared to non-semen-conditioned animals (week 8) (Fig. 1c). Our results indicate that semen exposure is associated with a 42% decrease in the risk of infection (Cox-regression $p$ = 0.039). To further demonstrate that reduced susceptibility is a consequence of semen alone, we compared Group 1 to 2 (Supplementary Fig. 4C) and Group 3 to 4 (Supplementary Fig. 4A) and found that SIVsmB7 did not impact susceptibility alone or when combined with semen (Supplementary Fig. 4; log-rank $p$ = 0.23 and log-rank $p$ = 0.94, respectively). However, comparison of Group 2 and 4, groups that only differed in the presence of semen during conditioning, demonstrated that semen conditioning resulted in reduced SIV susceptibility (Supplementary Fig. 4B, log-rank $p$ = 0.06). At completion of the 16-week low dose challenge, we transitioned all 11 uninfected animals to a 4-week high-dose repeated SIVmac251 challenge (Fig. 1b), where 10 of the 11 remaining animals became infected, confirming animals were still susceptible to infection. We did not detect significant differences in subsequent viral load, viral set-point, or CD4 decline post-infection between groups infected during low-dose period (Fig. 1e, f).

**Hormone levels did not impact rate of SIVmac251 infection.** Cyclical hormones, estrogen, and progesterone, have been shown to alter susceptibility to SIV through vaginal inoculation[13–15]. Hormones (progesterone and estradiol) and menstrual cycles were individually tracked throughout the study period in all animals (Supplementary Fig. 1 and Supplementary Table 2). For the 21 infections during the low-dose challenge period, we found near equal numbers of animals were infected in the luteal ($n$ = 11) and follicular ($n$ = 10) phases; moreover, we did not observe

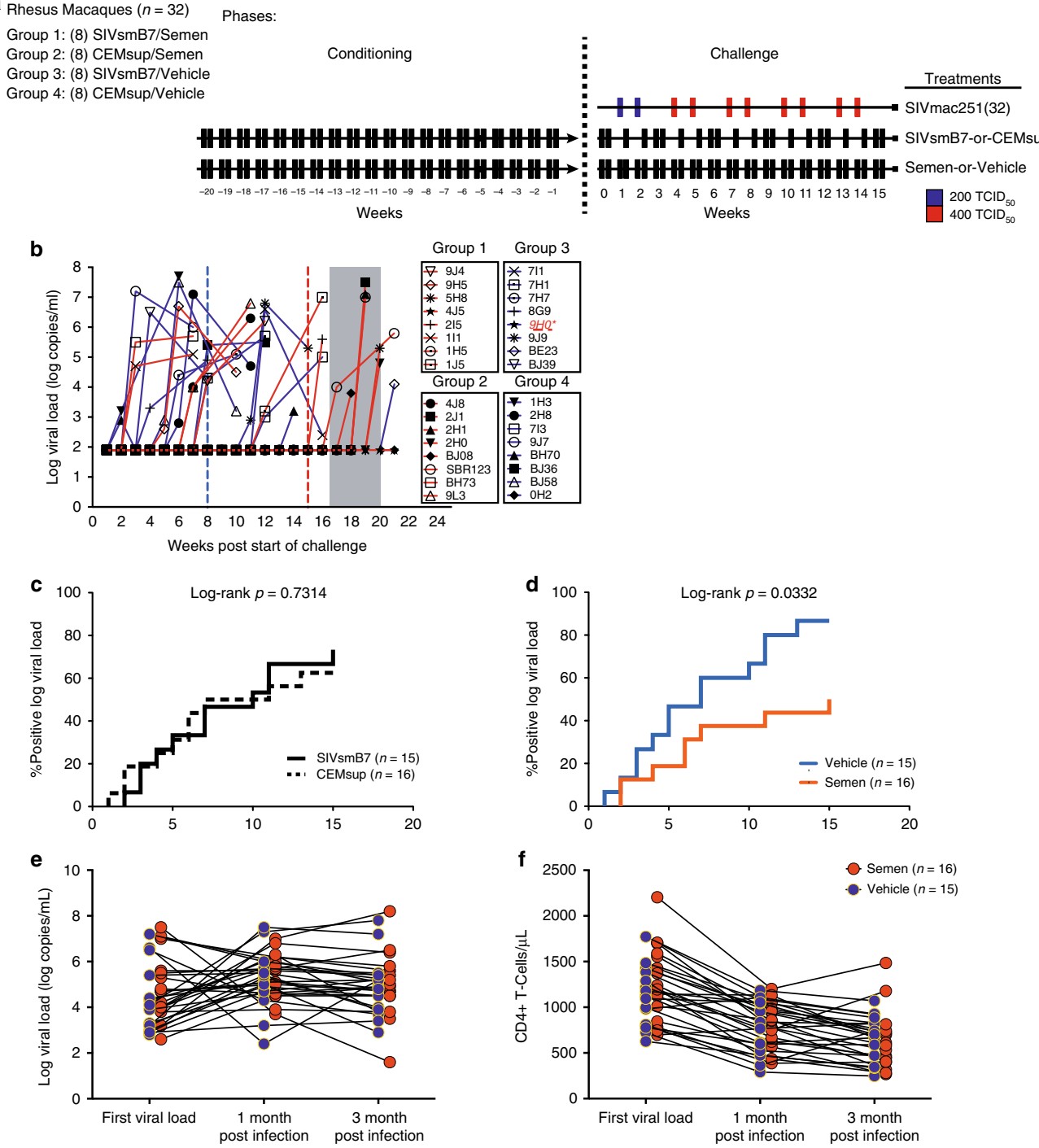

**Fig. 1** Intravaginal semen conditioning lowers susceptibility to SIV in rhesus macaques. **a** 32 Rhesus macaques (Macaca mulatta) were conditioned with intravaginal application of semen and/or SIVsmB7, a replication-incompetent SIV particle for 20 weeks. These animals were then challenged with low-dose SIVmac251 (200–400 TCID50) for 16 weeks intravaginally. **b** Log viral load line graphs for all 32 animals in the study with grouping listed. Blue vertical dashed line indicates time-point for 50% infection for non-semen condition groups and red vertical dashed line indicates 50% infection for semen conditioned group. **c** Survival curve comparing SIVsmB7 conditioned vs. animals receiving CEM_sup [Log-rank $p = 0.7314$]. **d** Survival curve comparing semen and vehicle-conditioned animals [Log-rank $p = 0,0332$]. 1- and 3-month post-acquisition VL (**e**) and CD4 (**f**) were similar in semen and vehicle-treated animals. Source data are provided as a Source Data file

any correlation between blood estrogen or progesterone levels and SIV susceptibility within our study.

**Chronic semen exposure alters circulating and tissue immune cells**. Studies of female sex workers[2] and Highly-Exposed Seronegative (HESN) women[16,17] suggest that a high frequency of

semen exposure may result in systemic changes, such as lower immune activation (i.e., CD38 expression) and lower CCR5 expression[18], which may correlate with HIV-1 resistance. Therefore, we measured the expression of activation markers and CCR5 in innate (NK, DC, macrophage) and T-cell subsets in peripheral blood before and after the 20-week conditioning

**Table 1 Biological data for animals used in low-dose SIVmac251 Challenge Study**

| Animal ID | Condition 1 | Condition 2 | Weight[b] | Age (Years)[b] | MHC I |
|---|---|---|---|---|---|
| 1J5 | Semen | SIVsmB7 | 6.14 | 4.68 | A08 |
| 1H5 | Semen | SIVsmB7 | 5.1 | 5.67 | ND |
| 1I1 | Semen | SIVsmB7 | 4.98 | 4.42 | A08 |
| 2I5 | Semen | SIVsmB7 | 5.08 | 4.65 | A08 |
| 4J5 | Semen | SIVsmB7 | 5.3 | 4.68 | A08/B01 |
| 5H8 | Semen | SIVsmB7 | 6.54 | 5.68 | B01 |
| 9H5 | Semen | SIVsmB7 | 6 | 5.42 | A01 |
| 9J4 | Semen | SIVsmB7 | 4.72 | 4.67 | A01/A08 |
| 9L3 | Semen | CEMsup | 6.62 | a | ND |
| BH73 | Semen | CEMsup | 5.3 | 4.58 | B17 |
| SBR123 | Semen | CEMsup | 4.54 | 5.42 | A08 |
| BJ08 | Semen | CEMsup | 5.46 | 4.59 | ND |
| 2H0 | Semen | CEMsup | 5.74 | 5.57 | A08 |
| 2H1 | Semen | CEMsup | 5.42 | 5.62 | ND |
| 2J1 | Semen | CEMsup | 5.94 | 4.61 | A01 |
| 4J8 | Semen | CEMsup | 7.1 | 4.66 | A01/B01 |
| 7H1 | No Semen | SIVsmB7 | 6.3 | 5.58 | A08 |
| 7H7 | No Semen | SIVsmB7 | 6.7 | 5.70 | A01 |
| 7I1 | No Semen | SIVsmB7 | 6.48 | 4.68 | A08 |
| 8G9 | No Semen | SIVsmB7 | 6.32 | 5.42 | A08/B01 |
| 9H0 | No Semen | SIVsmB7 | 6.74 | 5.66 | A08 |
| 9J9 | No Semen | SIVsmB7 | 6.3 | 4.65 | A08 |
| BE23 | No Semen | SIVsmB7 | 6.02 | 4.90 | B01 |
| BJ39 | No Semen | SIVsmB7 | 5.64 | 9.69 | A08 |
| BJ58 | No Semen | CEMsup | 5.54 | 4.17 | A08/B08 |
| 7I3 | No Semen | CEMsup | 6 | 4.60 | A01/B01 |
| 9J7 | No Semen | CEMsup | 6.36 | 4.42 | A01 |
| 0H2 | No Semen | CEMsup | 6.16 | 5.73 | B01 |
| 1H3 | No Semen | CEMsup | 7.42 | 5.61 | B01 |
| BH70 | No Semen | CEMsup | 6.34 | 4.91 | A02/A08 |
| BJ36 | No Semen | CEMsup | 5.92 | 3.46 | A08 |
| 2H8 | No Semen | CEMsup | 7.1 | a | A08/B01 |

Genotyping was done for Mamu-A*01, A*02, A*08, A*11, B*01, B*03, B*04, B*17, and B*08
[a]Unknown age
[b]Weight and age as of the start of challenge

period (Supplementary Table 3). Of interest, after conditioning, CCR5 expression on circulating CD4+ T-cells was significantly lower in semen-conditioned animals (Fig. 2a, b; Mann–Whitney $p = 0.0366$). CCR5 downregulation was observed in central memory CD4+ T-cells (Fig. 2b; Top Right Panel, Mann–Whitney $p = 0.0298$), with no difference in expression detected in the naïve (Mann–Whitney $p = 0.221$) or effector memory subsets (Mann–Whitney $p = 0.317$). Importantly, SIVsmB7 did not induce any changes in CD4+ CCR5 expression in any subset.

To determine if semen conditioning altered the tissue immune cell infiltrate in the macaque FRT, we repeated the same 20 week conditioning strategy as above (referred to as Groups 1a, 2a, 3a, 4a) in 14 new animals (3 animals per each non-semen group, 4 animals per each semen group); these animals were euthanized at the 20-week time-point, and immunohistochemistry was done on vaginal and cervical tissue (Fig. 3a). As prior work in macaques had shown that cervical immune infiltrates are relatively stable throughout the macaque menstrual cycle[19], we focused on interpreting ecto/endo cervical cellular infiltrate changes in naturally cycling females. Results showed clear differences between animals conditioned with semen when compared to those without semen (Fig. 3b–g). Animals in the semen conditioning groups had a higher number of CD4+ cells (Mann–Whitney $p = 0.0087$; 447.9 vs. 238 cells/mm$^2$), increased levels of both Ki-67 (Mann–Whitney $p = 0.0022$), and interferon-stimulated protein (ISP) Mx1 (Mann–Whitney $p = 0.0152$) within the submucosa of the ectocervix as compared to non-semen-conditioned animals (Fig. 3b–d).

Differences in the endocervical microenvironment were also detected. Endocervical MX1 expression was increased 6-fold (84563 vs. 13451 pixels/mm$^2$) amongst semen-conditioned animals (Fig. 3f). In contrast to the ectocervix, the endocervical compartment showed a more uniform increase in HLA-DR+ cells proximal to the columnar epithelium in animals receiving semen (Mann–Whitney $p = 0.0027$) (Fig. 3e). Enhanced HLA-DR levels correlated with increased recruitment of combined CD4+ T-cells and CD68+ myeloid cells (Supplementary Table 4; Mann–Whitney $p = 0.0006$ and 0.038, respectively). As work in mice has also shown that semen actively recruits and expands CD4 T-regulatory cells within the uterus[20], we compared levels of FoxP3 expressing cells within the endocervix of semen and vehicle-conditioned animals (Fig. 3g). FoxP3 expressing cells were significantly higher in semen-conditioned animals (Fig. 3g) (Mann–Whitney $p = 0.0427$; 16.40 vs 6.43 + cells/mm$^2$), and overall FoxP3 levels correlated with CD4 number in the endocervix (Fig. 3i and Supplementary Table 4). We further confirmed that these cells were T-cells by IF staining (Supplementary Fig. 8). By contrast, our analysis did not show significant differences in CD4+, CD68+, CD123+, or HLA-DR+ cell infiltrates in the ecto- or endocervix of animals conditioned over 20 weeks with SIVsmB7 as compared to CEM$_{sup}$ alone (Supplementary Figs. 5 and 6).

Consistent with prior reports of semen-induced expression of IFN-epsilon (IFNε) in cervical cells[21,22] and a sustained expression of IFNε observed within cervical epithelial cells in female sex workers (FSW) engaged in condomless sex[2], we found

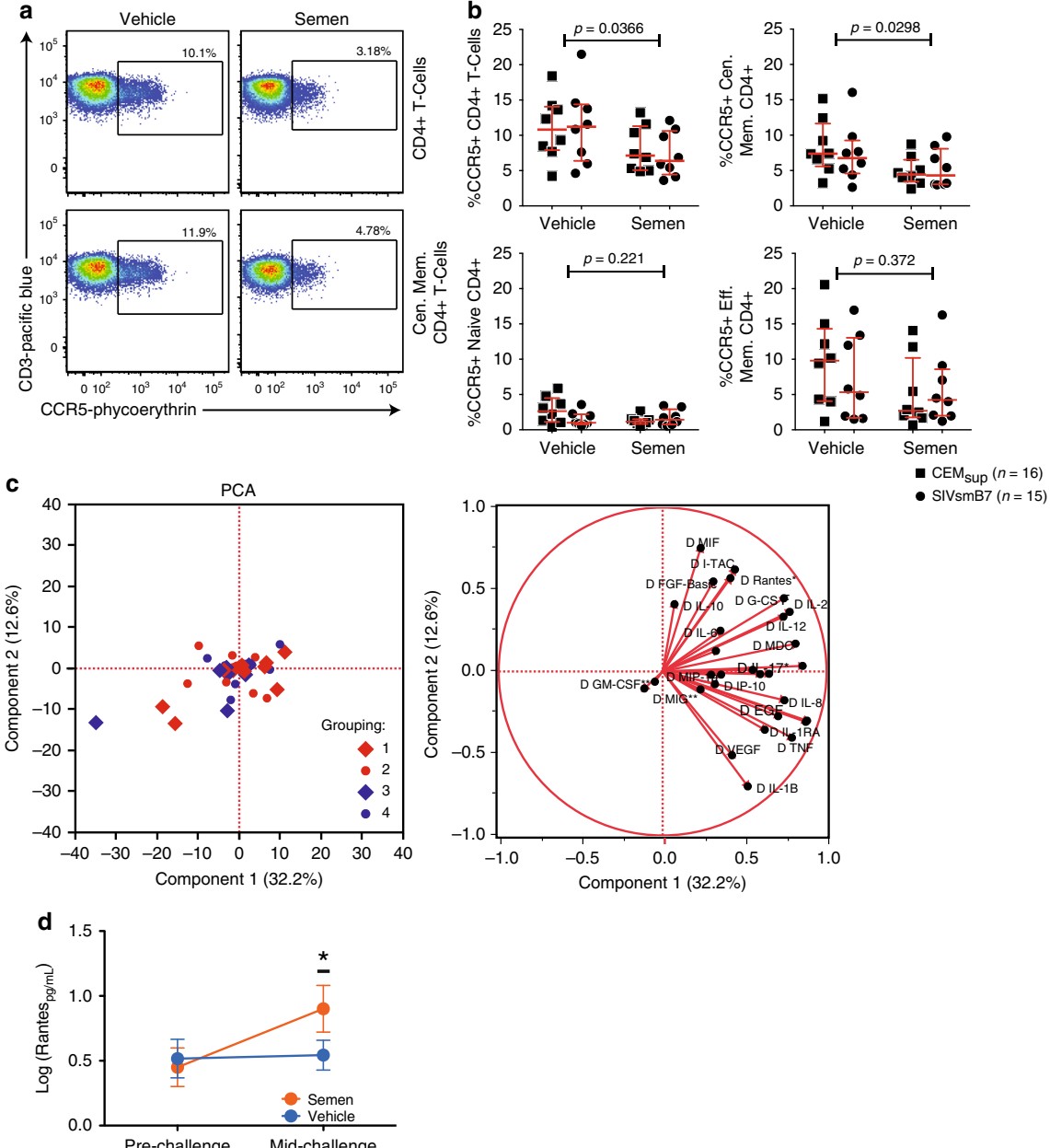

**Fig. 2** Semen conditioning lowers CCR5 expression on peripheral CD4+ T-cells. **a** Representative flow cytometry plots of CCR5 expression on peripheral CD4+ T cells in semen verses vehicle-conditioned macaques. **b** CCR5 expression on peripheral CD4+ T-cells was significantly lower amongst semen-conditioned animals (Man–Whitney $p = 0.0366$) (top left). Reduced CCR5 surface expression was confined to central memory CD4+ T-cells (Mann–Whitney $p = 0.0298$) (top right) as both naïve and effector memory CD4+ T-cells displayed no difference in CCR5 expression (bottom panels). **c** Principal Component Analysis (PCA) showed no segregation of animal groups based on vaginally secreted cytokines. **d** After the initiation of SIVmac251 low-dose challenge, semen-conditioned animals had an increase in RANTES. Error bars are represented as median with IQR. Source data are provided as a Source Data file

a trend of increased IFNε protein (Supplementary Fig. 7A) and mRNA (Supplementary Fig. 7B) expression within the reproductive tracts of semen-treated macaques. In agreement with reports showing IFNε to be a potent stimulus of Type I interferon regulated genes[23], Mx1 correlated with IFNε expression both in the endo and ectocervix (Fig. 3h, i, Supplementary Table 4, Spearman $p = 0.0171$ and Spearman $p = 0.0082$, respectively).

To address whether the introduction of human semen into the macaque FRT altered the immune activation state of the cervicovaginal compartment, we assessed the levels of 28 cytokines within the cervicovaginal fluid from all 46 studied macaques at baseline and after the 20-week conditioning period

(Figs. 1a and 3a). No significant difference in inflammatory cytokine levels was detected between semen-conditioned and non-semen-conditioned animals (Supplementary Fig. 9). Furthermore, Principal Component Analysis (PCA) of all cytokines tested also failed to reveal any distinct grouping associated with the conditioning treatment (Fig. 2c) to support a lack of inflammatory protein secretions before challenges started. To track cervicovaginal cytokine levels during the challenge period itself (for Groups 1–4), we redefined a baseline as the start of the challenge period and the endpoint, depending on the individual animal outcome, as either the time of positive plasma SIV RNA or the end of the low-dose challenge if uninfected. Results showed

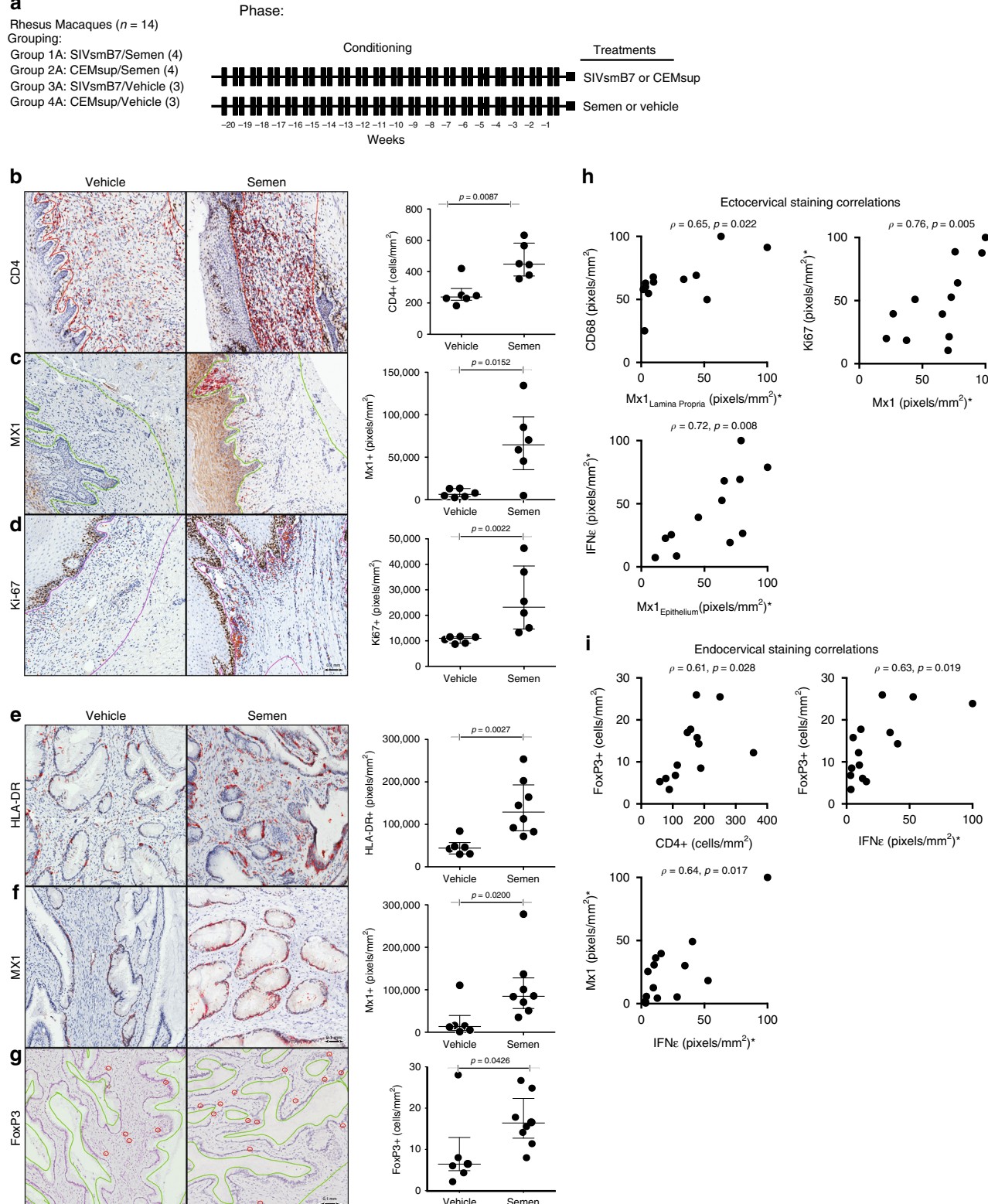

that semen-conditioned animals had higher levels of CCL5 (RANTES) when compared to non-semen-conditioned animals (Fig. 2d; $p = 0.0279$).

## Discussion

Taken together, our data demonstrate that repeated semen exposure results in lower levels of CCR5 on circulating CD4+ T-cells, higher MX1 expression in FRT tissues, increased number of cervical FoxP3+ T-regs, and elevated RANTES cervical secretions (upon challenge) all of which would support observations of a lower susceptibility against SIV vaginal challenge.

Two semen-induced mechanisms are consistent with the changes described.

First, IFNε expression induction by semen, through the estrogen receptor in epithelial cervical cells[24], could account for the MX1 induction observed. Although engagement of the

**Fig. 3** Semen increases CD4+ cell infiltration and Mx1 expression within the cervix. **a** To assess the impact of semen on cellular infiltrates of the female reproductive tract, 14 macaques were euthanized after undergoing the same 20-week treatment sequence as the 32 animals that underwent challenge. For clarity, these groups are designated with an "A" following the original group name. The ectocervix displayed 3 distinct changes in response to semen pretreatment (analyzed as vehicle $n = 6$; semen $n = 6$). **b** semen-conditioned animals had nearly twice as many CD4 as semen-naïve animals (447.9 vs 238 cells/mm$^2$; $p = 0.0087$). **c** These animals also had significantly higher levels of Mx1 staining as compared to semen-naïve animals which either had little or no Mx1 staining ($p = 0.0152$). **d** Similar to Mx1, Ki-67 within the lamina propria was significantly higher in semen-conditioned animals with most staining occurring near the basal epithelium. The endocervix displayed three distinct differences between semen and vehicle-conditioned animals (analyzed as vehicle $n = 6$; semen $n = 8$). **e** HLA-DR staining was 3-fold higher in semen-conditioned animals ($p = 0.0027$). **f** Mx1 staining of the ectocervix mirrored staining found in the ectocervix with vehicle-conditioned animals having little to no staining in comparison ($p = 0.0200$). **g** Semen-conditioned animals had nearly 3× more FoxP3+ cells within their endocervices as compared to vehicle-conditioned animals ($p = 0.0426$; 16.4 vs 6.43). **h** Select significant ectocervical correlations are shown for CD68 vs Mx1 (lamina propria); Ki67 vs Mx1; and IFNε vs Mx1 (epithelium) are shown. **i** Select significant endocervical correlations are shown for FoxP3+ vs CD4+ cells; FoxP3+ cells vs IFNε; and Mx1 and IFNε. Brown (Hex Code#613026/sensitivity 40 or 60) IHC staining was pseudo-colored red (Hue: −28, Saturation: +75) for visual clarity. Quantified tissue regions are outlined. Mann–Whitney $U$ testing was used to compare groups. $P$-values below 0.05 were considered significant. Error bars are shown as median with IQR. Spearman's rho was used to determine the correlation between staining parameters. * Indicates staining values were normalized to max. Following shipment from the study site, it was determined that one animal from Group 5 and one animal from Group 6 had endocervical tissues too degraded for interpretable staining. All figures shown with 0.1 mm scale bar included. Source data are provided in Source Data file

estrogen receptor by semen factors has not been directly shown, estradiol is a component in male semen[25].

Second, high levels of TGF-β[20] are present in semen with the potential to induce TGF-β signaling and expansion of local T-regs[20]. Importantly, supporting the effects of human semen on macaque FRT, recombinant human TGF-β has been reported to modulate macaque T-regs in vitro[26,27]. Furthermore, in an independent cohort of HESN female commercial sex workers with long-term high-level intravaginal semen exposure, elevated levels of both T-regulatory and tolerogenic myeloid cells have been described within their cervicovaginal mucosa[28]. Our data now provides a direct link between these collective observations in humans and semen-mediated effects on increasing T-regulatory phenotypes as evidenced in non-human primates. Taken together, data suggest that a tolerogenic vaginal environment can be induced by repeated semen exposures which may lower susceptibility for a productive SIV infection[29,30].

Importantly, the mechanisms mediating the observed SIV resistance are overwhelmed by increasing the challenge dose, indicating that semen exposure can modulate infectivity at low infection doses but does not block infection if viral challenge dose increases.

Prior work in HESN female sex workers demonstrated that a break from sex work increased the risk of HIV infection upon return to sex work[31]. Our data suggest that any protective mucosal modulation induced by semen exposure may be reduced during a sex work hiatus potentially increasing susceptibility to HIV-infection upon resuming exposure. However, a limitation of our data is that it does not address the impact of other important factors such as co-infections, altered microbiome, seminal exosomes[32], or sustained FRT inflammation on semen-induced changes. For example, it remains unknown if repeated semen exposure modulates pre-existing inflammatory states in the FRT, which have been associated with increased susceptibility to infection[32]. This study also does not address the infectivity of cell-associated SIV/HIV within semen[33,34], nor does it address whether repeated semen exposures from HIV viremic or anti-retroviral therapy suppressed men has similar effects with regards to the added presence of anti-HIV Ig and cytokine/chemokine levels (especially of RANTES) when compared to semen from uninfected men[4,35]. Finally, immune responses against xenogeneic proteins (within CEM inoculum or semen) were not directly tested. Although these can not be excluded from contributing to a tolerized state, we did not detect indirect evidence of increased xenogeneic immune responses which could account for direct antiviral responses based on (a) the same human-derived CEM proteins administered to all animals, (b) the lack of an enhanced FRT inflammatory state in the semen-treated group as compared to the non-semen-treated group, and (c) the absence of an "adjuvant" effect resulting in increased anti-SIV responses in semen-conditioned groups before infection.

Importantly, the SIV infection models used to test candidate vaccines currently do not consider the effects of repeated semen pre-conditioning, failing in this to represent the conditions observed in sexually active women having regular to high frequency of condomless sex. Our data further indicate a need for greater attention to the impact of sustained high frequency semen exposures in human clinical studies as a variable factor that may signal for a lower infectivity in female participants selected for HIV prevention studies. For example, it is common practice that all human anti-HIV microbicide or HIV vaccine clinical trials target sex workers engaged in condomless sex which our data would indicate may include a subset of women with inherent lower HIV infectivity states due to sustained rates of semen exposure. Independently of added factors (e.g., trauma, coinfection), our study identifies a role for the frequency of intravaginal semen exposure as an independent factor able to lower the intrinsic susceptibility of heterosexual transmission of HIV-1.

## Methods

**Animal care and experimental procedures**. Forty-six Healthy female rhesus monkeys (*Macaca mulata*) were acquired from the Caribbean Primate Research Center (CPRC) of the University of Puerto Rico (UPR)—Medical Sciences Campus (MSC). Animals were quarantined for six months and maintained at the AAALAC-accredited facilities of the Animal Resources Center, UPR-MSC. All animal studies were approved by the UPR-MSC, Institutional Animal Care and Use Committee (IACUC), and comply with the Guide for the Care and Use of Laboratory Animals. Animal Welfare Assurance Number: A3421, Protocol number: 3380113. Before experimental procedures, animals were single housed, as approved by the IACUC. All procedures were conducted under anesthesia (intramuscular administered Ketamine, 10–20 mg/kg). Blood samples were taken every other day to assess estrogen and progesterone levels prior to and during the experimentation. A trained veterinarian at the Animal Research Center provided continued monitoring.

Adult female rhesus macaques were randomized into 4 groups for a twice/week conditioning over 20 weeks (detailed below) before the start of low-dose infectious SIV challenges under the following groups: Semen/SIVsmB7Control CEM$_{sup}$ (Group 1), Semen/Control CEM$_{sup}$ (Group 2), No Semen/SIVsmB7/Control CEM$_{sup}$ (Group 3), No Semen/Control CEM$_{sup}$ (Group 4) as outlined in Supplementary Tables 1 and 4. Animals analyzed at 20 weeks of conditioning without proceeding to a challenge period are referred to in figures as Groups 1a, 2a, 3a and 4a, respectively. Both semen and semen-naïve animals were on average 5 years of age. Despite randomization, semen-treated (mean: 5.7 kgs) and semen-naïve (mean: 6.4 kgs) groups did have modestly different weights.

Conditioning and SIV challenge intravaginal inoculum was a total volume of 500 μl deposited atraumatically in front of the cervix of recumbent anaesthetized animals. Conditioning inoculum given to all animals with in the study consisted of

one of the following (a) RPMI 1640, (b) pooled human semen (30% volume in medium), (c) 500 ng p27 of replication-incompetent SIV E660 (SIVsmB7[12]), or (d) combined SIVsmB7 and pooled human semen. As detailed in results text and in Supplementary Table 1, inoculums included supernatant from control CEM cells (CEM_sub). Conditioning inoculum was given as detailed in Supplementary Table 1/ Supplementary Fig. 2. SIV challenge following the 20-week conditioning period was performed in 32 animals (8 animals per group, Supplementary Table 1) over a 16-week low-dose intravaginal challenge with SIVmac251 (200–400 TCID$_{50}$) (Fig. 1a and Supplementary Table 1). The SIVmac251 challenge with or without semen was performed in place of the first of the two weekly conditioning treatments and continued until SIV RNA was detected in the blood. A rest period of one week wherein no challenge was given was done after every 2 weeks of challenge. Once the infection was established animals were monitored for 3 months for establishment of a set-point plasma viral load (PVL). After SIV infection was confirmed, PVL was measured at a minimum at 1 month and at 3 months post-infection prior to euthanasia. Animals not challenged were euthanized immediately following the 20-week preconditioning. The semen-naïve animal 0H2 was euthanized after lack of infection was confirmed following its high-dose challenge.

Euthanasia was performed only on fully anesthetized animals by injection of Pentobarbital Sodium at 390 mg/ml; 1cc/10lbs IV. Vaginal and cervical tissues were taken for analysis.

**Immuno-histochemistry.** Tissues (only from groups 1a, 2a, 3a, and 4a animals) were deparaffinized and rehydrated in deionized water. Heat-induced epitope retrieval was performed using the water-bath method (95–98 °C for 10–20 min) in 10 mM sodium citrate, pH6.0 for CD123 detection (Sc-681, 1:2000, Santa Cruz Biotechnology Inc.), and Mx1 (1:3000, ProteinTech) and CD68 (KP1, 1:200, Dako). For CD4 detection (IF6, 1:60, Leica Microsystems), epitope retrieval was performed using high-pressure Decloaking Chamber (121 °C for 35 sec, Biocare Medical) in 1 mM EDTA, pH 8.0, followed by cooling to room temperature. Tissue sections were blocked with SNIPER Blocking Reagent 5% Non-fat milk (Biocare Medical) for 1 hr at room temperature. Endogenous peroxidase was blocked with 3% (v/v) $H_2O_2$ in methanol TBS (pH7.4) 3% [v/v] for CD68, Mx1 and CD123, and 0.9% for CD4 detection). Primary antibodies were diluted in 10% SNIPER Blocking Reagent in TNB blocking buffer (Tris-HCl, pH7.5, 0.15 M NaCl, 0.05% Tween 20 with 0.5 Dupont blocking reagent buffer) and incubated overnight at 4 °C. After the primary antibody incubation, sections were washed and then incubated with mouse, goat, or rabbit polymer system reagents conjugated with either horseradish peroxidase or alkaline phosphatase (ENVISON kit; Dako) according to the manufacturer's instructions, and developed with 3,3′-diaminobenzidine (Vector Laboratories). Sections were hematoxylin counterstained, mounted in Permount (Fisher Scientific), and examined by light microscopy. Non-specific, IgG was used as isotypic control.

Sections of each stained slide were digitized using Scanscope (Aperio), the image was opened in ImageScope, and endocervical areas were selected with ImageScope drawing tools for analysis. CD4+, CD68+, Mx1+ or CD123+ cells were quantified by using a positive pixel count algorithm in the Spectrum Plus analysis program (Version 9.1, Aperio). The parameters of the algorithm were manually tuned to match the specific staining markup image accurately over background DAB stain. Once the parameters were set, the algorithm was applied automatically to all digital slides to measure the number of cells of interest. Data were reported as positive staining cells per square millimeter. Any absent tissue staining data in specific animals was due to poor quality blocks (all available data was used).

**Cervicovaginal fluid collection and recovery.** Weck-Cel sponges (Beaver-Visitec, Waltham MA) were pre-wet with sterile PBS. Sponges were then inserted gently into the vaginal space and used to collect sample from the cervical os. Sponges were then stored at −80 °C until use. Sponges were then thawed and samples were eluted using a microcentrifuge.

**Binding antibody multiplex assay.** SIV binding antibody multiplex assay (SIV-BAMA) was used to measure concentrations of IgG/IgA specific to SIV$_{mac251}$ gp130, SIV$_{smE660}$gp140, SIV$_{mac32h}$ gp140, SIV$_{mac239}$gp120, and SIV$_{smE543}$gp120 from vaginal weck-cel elutions using a custom SIV multiplex ELISA[36]. SIV antigens were coupled to carboxylated fluorescent beads and incubated with samples diluted 1:2. SIV-specific Ig were detected with biotinylated goat anti-monkey IgG or IgA, followed by incubation with streptavidin-phycoerythrin (PE). Beads were washed, and data were acquired on a Bio-Plex instrument to measure florescence. Positive and negative monkey serum controls were used in each assay. Specific binding activity values, calculated as MFI × dilution/total IgG concentration (nanograms per milliliter), are reported. Positivity criteria included reactivity to two or more antigens with values 3-fold over the baseline visit and the cutoff was established using negative samples. The total IgG concentration in mucosal samples was measured by a custom ELISA after sample elution and preparation for binding antibody assays.

**Viruses.** SIVmac251 was diluted from a 20,000 TCID$_{50}$/ml viral stock grown in ultra-SPF rhesus macaque PBMC produced by Dr. Ron Desrosiers (New England

National Primate Research Center, Harvard Medical School) and kindly provided by Dr. Nancy Miller (NIAID) through contract #N01-AI-30018. Viral stock was previously titrated in female macaques to determine appropriate low dose challenge start point.

SIVsmB7 is a virus-like particle (VLP) derived from a clone of a CEMx174 cell line stably infected with SIVsmH3. SIVsmB7 is non-infectious due to a 1.6 kbp deletion including integrase, vif, vpr and vpx genes. Cell-free SIVsmB7 and CEMx174 supernatant (CEM mock control) were isolated by standard 20% (w/v) sucrose gradient ultracentrifugation. P27 ELISA was used to determine 500 μg P27 SIVsmB7 dose. CEMx174 dose was established by equal protein quantification with SIVsmB7 dose.

**Semen preparation.** De-identified, discarded cryopreserved semen samples were received under IRB exemption from the Andrology Laboratory of Penn Fertility Care, University of Pennsylvania. 30 individual vials of semen were thawed at 37 °C and pooled into a single master pool mix before refreezing into one-time use aliquots and stored at −80 °C until rethawed at 37 °C on the day of use.

**Cytokine multiplex assay.** Life Tech Non-Human Primate Cytokine 28-plex Assay (Carlsbad, CA) was used to measure cervicovaginal cytokine levels according to manufacturer protocol. Data was acquired using a Bio-Plex 200 System (Biorad, Inc.). Cutoffs were determined as 50% of the MFI of the lowest point on the standard curve for a given cytokine.

**Flow cytometry.** Archived PBMC were thawed in 37 °C water bath and washed. Cells were then stained with Aqua Live/Dead (Life Tech). After washing, cells were stained with 3 different staining panels (Supplementary Table 4) to explore CD4+ T-regulatory cell (Supplementary Fig. 10), T-Cell, NK-Cell (Supplementary Fig. 11), macrophage and Dendritic cell (Supplementary Fig. 12) activation. Cells were fixed with 4% PFA and collected on LSR II (BD Bioscience). A minimum of 75,000 live lymphocyte events was collected from each sample. Subsequent gating was determined from FMO and/or isotype staining.

**DNA isolation and MHC genotyping.** Genomic DNA was isolated from a maximum of $3.0 × 10^6$ peripheral blood mononuclear cells using the MagNA Pure LC system (Roche Applied Science) and the MagNA Pure LC DNA Isolation–Large Volume protocol (version 3.0) according to manufacturer's guidelines. The elution volume of extracted DNA was 200 μl of MagNA Pure LC DNA Isolation–Large Volume elution buffer. DNA concentrations (ng/μl) and Abs 260 nm/Abs 280 nm ratios were determined using a NanoDrop UV Spectrophotometer (NanoDrop Technologies). Genotyping for Mamu-A*01, A*02, A*08, A*11, B*01, B*03, B*04, B*17, and B*08 as previously described[37–39]

**The quantification of IFN-ε expression in FRT tissues using qRT-PCR.** Macaque FRT tissues were homogenized with a power homogenizer in TRIzol solution (Life Technologies), followed by purification with an RNeasy Mini Kit (Qiagen, Hilden, Germany). The cDNA was synthesized from 5 μg total RNA using an Oligo (DT) primer and Superscript III RT (Life Technologies). qRT-PCR was conducted in a final volume of 20 μl with 800 ng cDNA, 0.2 μM of each primer, and Platinum *Taq* High Fidelity Polymerase (Invitrogen) using the CFX96 Real-Time detection system (Bio-Rad Laboratories, Hercules, CA, USA), using a hot start (95 °C for 3 min) and 40 amplification cycles (95 °C for 15 s, 57 °C for 30 s). The following primers and probes were used for amplification and detection: Rh-IFN-ε forward CTC TTG AAT AAG TTG CAA ACC TCA and Rh-IFN-ε reverse 5′-TCT GCT GAA GCA TCT CAT GG-3′; GAPDH forward 5′-ACA TCA TCC CTG CCT CTA CT-3′, Rh-IFN-ε probe 5′-/56-FAM/AGA AGT CTT /ZEN/TGA GTC CTC AGC AGT ACC A/3IABkFQ/−3′; GAPDH probe 5′-/56-FAM/CAA GGT CAT/ZEN/ CCC TGA GCT GAA CGG/3IABkFQ/−3′.

**Hormone measurements.** Blood was taken every other day throughout the menstrual cycle of the macaques under study.

Estradiol and progesterone concentrations were measured by enzyme-amplified chemiluminescence (Immulite 1000, Siemens). The analytical limits of sensitivity of the estradiol and progesterone assays were 15 pg/mL (references range of 20–2000 pg/mL) and 0.1 ng/mL (reference range of 0.2–40 ng/mL), respectively.

**CD4 and CD8 counts.** For tracking of macaque CD4 and CD8 counts for both the viral stock titration and Phase II of the study, TruCount Absolute Count Kits (BD) were used according to manufacture protocol. Briefly, peripheral blood was labeled with CD3e, clone SP34, (BD, Cat#556611), CD8, clone SK1, (BD, Cat#347314) and CD4, or clone L200, (BD, Cat#551980). After labeling, red blood cells were lysed and washed. Labeled cells were collected with a BD FACSCalibur Flow Cytometer and CD4 and CD8 counts were assessed.

**SIV Viral loads.** Plasma samples were spiked with armored RNA (aRNA; Asurgen) and centrifuged at 25,000×$g$ for 1 h. Viral RNA (vRNA) was extracted from the pellet with Proteinase K (2.5 μg/μl; Life Tech) and the High Pure Viral RNA kit

(Roche). Eluted vRNA (100 μl) was then subjected to the RNA Clean and Concentrator kit (ZYMO Research) and eluted in 50 μl, from which 15 μl were reverse transcribed using MultiScribe™ Reverse Transcriptase (Life Tech) in a 50-μL gene-specific reaction. Fourteen microliters of cDNA were added to TaqMan gene expression master mix (Life Tech), along with primers and a probe targeting the gag region of SIVmac251, and subjected to 40 cycles of qPCR analyses. Fluorescence signals were detected with an Applied Biosystems 7900HT Sequence Detector. Data were captured and analyzed with Sequence Detector Software (Life Tech). Viral copy numbers were calculated by plotting CT values obtained from samples against a standard curve generated with in vitro-transcribed RNA representing known viral copy numbers. The limit of detection of the assay was five copies per reaction volume or 40 copies per ml of plasma.

**Statistical analysis**. Shapiro-Wilk tests were carried out to detect normality distribution of variables and then to determine appropriate statistical tests or procedures. Survival analysis was done by Log-rank testing. Differences between groups were tested using Wilcoxon rank-sum or Student's t-test depending on the outcome normality assessment. P-values less that 0.05 were considered significant. Multi-testing correction was not done.

**Reporting summary**. Further information on research design is available in the Nature Research Reporting Summary linked to this article.

## Data availability
All data is available in the main text or the supplementary materials. The source data underlying Figs. 1b–e, 2b–d, 3b–i and Supplementary Figs 1, 3A–B, 4A–C, 5, 6, 7A–B, and 9 are provided as a Source Data file.

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

## Acknowledgements
We would like to thank the Institute of Molecular Virology, Ulm University Medical Center, Ulm, Germany and the Penn Medicine Division of Reproductive Endocrinology and Infertility for providing the semen samples used in this study. We acknowledge critical reviews of this manuscript by Drs. Roger Le Grand (Univ. of Paris-Est, IDMIT) and Michaela Muller-Trutwin (Pasteur Institute). This work was funded by NIH grants R01 AI084142 and R01 AI094603 to E.K. and L.J.M., NIH grant P40 OD012217 to M.I. M., and T32 AI070099 to S.A.A. Additional support was provided by The Philadelphia Foundation (Robert I. Jacobs Fund), Kean Family Professorship, the Penn Center for AIDS Research (P30 AI 045008), Cancer Center Grant (P30 CA10815), Duke Center for AIDS Research (5P30 AI064518), and the DFG (CRC 1279, MU3115/8-1 and SPP 1923). The funders had no role in study design, data collection and analysis, decision to publish, or preparation of the manuscript.

## Author contributions

Investigation and methodology by S.A.A., M.M., G.K., I.V.R., S.M.N., D.B., J.J., L.A., X.Y., M.W., D.W., Q.L., A.F., J.M., F.K., C.C., G.D.T., C.S., P.A.M., Q.L., E.K., and L.J.M.; Formal analysis by S.A.A., A.F., X.Y., and QL; Writing original draft by S.A.A. and L.J.M.; Writing-review & editing by S.A.A., M.M., G.K., I.V.R., S.M.N., D.B., J.J., L.A., X.Y., M.W., D.W., Q.L., A.F., J.M., F.K., C.C., G.D.T., C.S., P.A.M., Q.L., E.K., and L.J.M. Supervision: L.J.M.

## Additional information

**Competing interests:** The authors declare no competing interests.

