## [Peer Review File · Nature Communications]

Reviewers' Comments:

Reviewer #1:

Remarks to the Author:

In this manuscript Abdulhaqq et al., investigate whether the repeated exposure of female reproductive tract (FRT) to the semen could decrease the SIV acquisition in non-human primate model. They have shown that the repeated exposure of semen resulted in lower CCR5 expression in circulating CD4 T-cells, increased infiltration of CD4+ T-cells to the cervicovaginal mucosa and increase expression of Mx1 and FoxP3 in FRTs. The paper is very well written and have appropriate control groups. Study is very interesting and important for understanding the HIV transmission / pathogenesis and better evaluating HIV vaccine/ preventive studies. However, there are some major concern about the experiment design and interpretation of results.

Major points:

The main claimed of this paper is that the repeated exposure of FRT to semen will reduce SIV acquisition significantly. Thus, they should compare the acquisition rate between semen /CEM (gp2) vs vehicle/CEM (gp4). Instead, they have mixed groups according to the exposure to the smB7 or semen (Fig.1B, C) and claimed that the semen can reduce SIV acquisition compared to vehicles. In the Fig. S4 A and B; They also compare Control /CEM group with Control /smB7 group. Is this control group similar to vehicle group? This is so confusing and need a clarification. Also, they should clearly explain why preconditioning needed for this study.

The effect of semen in the peripheral blood CD4 T cell is very significant finding in this study. However, In the Fig 2.B each group looks like mixture of a semen treated and untreated group. Can you have separated smB7 exposed animals from non-exposed animals in each group? It is important to see the significant difference is not a result of exposure to smB7. Didn't you study the reduction of Ki67+CD4 T cells in the blood in the animals exposed to semen? haven't you study the Immune cell responses in lymph nodes (draining lymph nodes of FRT) of sacrificed animals?

Immunohistochemistry data very convincing and shows that the effect of semen in tissue CD4 T cells and MX1 and Ki67 expression. However, sample numbers used in these graphs are not consistent. Fig 3 B, C, D: 6 animals in each group. In the Fig. 3 E, F, G: no-serum group has 6 and semen group has 8. Why this inconsistency. very confusing? The results of these tissues will be enhanced if they have stained for CXCR3 or IP10 to demonstrate the inflammatory status of the tissues. Have you stained for NK or macrophages, DC markers in the vaginal tissues? Authors discuss tissue associated immune mechanisms such as ADCC, so it is important to study tissue specific immune cells in addition to T cells (by flow cytometry or immune histochemistry).

Minor points:

Fig. 1 D and E. It is important to see how each animal viral load and CD4 counts changes during the infection in each group. Thus, please present data in line charts.

Page 5: line 40 "Taken together, our data demonstrate that repeated semen exposure results in higher FRT tissue MX1 expression, lower levels of CCR5 on CD4+ T-cells" have you shown CCR5 level on tissue specific T cells?

Although not studied it is important to mention about the possible contribution of vaginal microbiome for the FRT immunological responses during semen exposure.

Reviewer #2:
Remarks to the Author:

The manuscript by Abdulhaqq et al explores the ability of semen to alter the vaginal mucosa and impact SIV transmission. The study makes innovative use of the SIV macaque model to accommodate repeated semen exposure occurring in women at risk of HIV acquisition. The study identifies a substantial reduction in risk of SIV acquisition in the semen-conditioned macaques and goes on to explore some relevant mechanisms by which protection is provided. The findings are innovative and have the potential for having a high impact with regard to heterosexual transmission of HIV, however there are some major issues that dampen the overall enthusiasm for the manuscript in its current form.

Major Points:

" First is the question of whether the effects observed are semen specific, or alternatively are the effect of human semen in a macaque vagina. In other words, whether cross species immune responses are responsible for the altered immune responses and impact on SIV transmission. One previous finding that makes this concern particularly relevant is the vaccine studies from the early 1990s that demonstrated that a vaccine being produced in human cells was protective against a challenge SIV virus produced in human cells. However, it turned out that the protective immune responses were actually to human proteins, not SIV proteins. Two particular issues should be addressed.

Number 1, the authors hypothesize that TGF-beta and prostaglandins in semen might influence levels of local T-regs in the vagina(line 183). Is it known if human versions of these proteins can influence macaque T-regs?

Number 2, if there are immune responses being generated to human proteins or sperm then these may be indicative of a local immune environment in the vagina that is being modulated by human proteins. Assessing for anti-human sperm or anti-human antibodies would be one way to address this issue.

" Second, the manuscript is difficult to follow with regard to which group is getting which of the conditioning in their vaginas. Therefore, the suggestion is to better specify the different groups with some type of nomenclature that one can easily follow. For example, groups A, B , C, D could be used to designate the macaque groups that were SIV challenged (per figure S2 and table S1). In addition, a group designation of E, F, G, H could be utilized for the macaque groups that were not SIV challenged. Then for example, Figure 1 instead of trying to riddle which groups were included in the analysis or 'vehicle' or 'semen' it could say specifically which groups were evaluated.

" Third, while assessing peripheral CCR5 CD4 cells is interesting, it would be even more interesting to quantifying the levels of SIV/HIV target cells in the cervix or vaginal area where the semen is being placed.

" Fourth, the discussion and data addressing the potential mechanisms by which the semen might be achieving the SIV transmission reduction could be discussed in more detail (with less text designated for discussion of how the findings impact HIV studies).

Minor Points:

" Table S3 and figure 2 describe opposite findings. Figure 2 has the CCR5 lower in the semen treated macaques and table S3 has the CCR5 lower in the semen-naïve.

" There have been observations that sex workers that take a break have an increased rate of HIV

seroconversion soon after they return to work. I think these observations are important for the reasoning and understanding of why your experiments are relevant. I am not exactly sure the reference for this, however there is the manuscript by Plummer and colleagues (Kaul, Rowland-Jones et al JCI 2001) that does assess sex workers taking breaks which provides some insights. These findings should be described.

" Please define the term "high-level semen"

" While the study provides clear evidence of FRT immune environment following semen exposure, it is over-speculation to suggest here that semen exposure could impact ADCC or vaccine protection outcome. Suggest limiting discussion to logical extrapolation of findings described.

" While the data is generally presented with clarity (except as discussed above) the quality of the writing in manuscript is lacking. Spending some time refining the narrative would benefit the readers.

Reviewer #3:

Remarks to the Author:

COMMENTS TO THE AUTHORS

This manuscript entitled "Repeated semen exposure decreases cervico-vaginal SIVmac251 infection in rhesus macaques" by Dr. S. A. Abdulhaqq and colleagues describes results of a very important study that has implications for vaccine design against HIV. Overall, the protocol was difficult to follow and critically evaluate and details for parts of the study were missing. For sake of simplicity, I would suggest a simple change in nomenclature. The animals administered SIVsmB7 could be abbreviated to antigen (Ag) and those administered semen abbreviated to (Se). Thus, there would then be 4 groups of animals, group 1 Se+/Ag+, group 2 Se+/Ag-, group 3 Se-/Ag+ and group 4 Se-/Ag-- throughout the text OR something along these lines.

Basically, the study design was aimed at duplicating the study performed on female sex workers in Kenya in which such sex workers were found to be relatively resistant to HIV infection despite the fact that they were repeatedly exposed to HIV via exposure to semen from HIV-1+ men. The authors set up a very well designed study in which groups of female rhesus macaques were conditioned with (i) group 1 (n=12), a 30% suspension of pooled human semen (Se) + 500 ng p27 of a replication incompetent SIVsmB7 (Ag) (ii) group 2 (n=12), Se + control fluid (iii) group 3 (n=11), control for Se + Ag and (iv) group 4 (n=11), Se-/Ag-. Thus, for simplicity, the 4 groups consisted of Se+/Ag+, Se+/Ag-, Se-/Ag+ and Se-/Ag-. The animals were conditioned with the administration of the appropriate combination of SE and Ag twice a week for 20 weeks. Following the conditioning regimen, 8 animals from each group were subjected to low dose repeated challenge with SIVmac251 starting at 200 TCID50 (week 1/2), 400 TCID50 (weeks 4 and 5; 7 and 8; 10 and 11; and weeks 12 and 14), 800 TCID50 (weeks 16 and 17) and 3200 TCID50 at weeks 19 and 20. The animals were rested for 1 week between each challenge dose. Once the monkeys were infected, the challenges were discontinued.

The authors first of all found that there was no difference in the susceptibility of animals that were conditioned with Ag (groups 1 + 3) versus those that were not (groups 2 +4). However, those that were conditioned with Se (groups 1 + 2) showed significant level of resistance to infection as compared with animals that were not conditioned with Se (groups 3+4). There was, however, no differences in either viral loads on the first positive sample or those at 1 or 3 months p.i. or rate of CD4+ T cell decline between the groups of animals suggesting that once infected they appeared to be similar for viral loads or CD4 counts. The authors examined levels of CCR5 expressing cells CD4+ T cells in the blood and found that the Se+ animals had a marked decrease in CD4+/CCR5+ CM but not naïve or EM cells as compared with the Se- animals in samples taken at 20 weeks post conditioning as compared with baseline. The authors next examined levels of 28 cytokines in the CVL again in samples collected at 20 weeks post conditioning versus baseline and found no difference between Se+ and Se- animals except for the levels of RANTES that were higher in Se+ versus Se- animals.

The authors next performed IHC studies of vaginal and cervical tissues of a new set of 4 Se+/Ag+, 4 Se+/Ag- and 3 Se-/Ag+ and 3 Se-/Ag- animals that were similarly conditioned and sacrificed at 20 weeks post conditioning. The authors found increased numbers of CD4+ that also expressed Ki67 and Mx1 within the submucosa of the ectocervix of the Se+ animals as compared with Se- animals. Increase of Mx1 was correlated with increased levels of interferon epsilon. There was also increased levels of HLA-Dr + cells and FoxP3+ cells in the Se+ versus Se- animals.

The authors thus conclude that semen conditioning of the animals conferred a state of decreased susceptibility to infection that was associated with decreased frequencies of CCR5 expressing CD4+ T cells, increased numbers of FoxP3+ Tregs and elevated levels of RANTES and these therefore could be the potential mechanisms for resistance of HIV infection in the sex workers in Kenya.

As stated above, this is a well performed study that DOES provide a very useful platform to carry out further studies on the detailed mechanisms involved in the resistance mechanisms provided by semen pre-conditioning. There was clearly a lot of work performed and involved a significant number of nonhuman primates and particularly female rhesus macaques that are difficult to obtain and the authors are commended for their effort. There are a number of issues that the authors, however, need to address:

1. It seems that would clearly be some sort of xenogeneic response to human proteins including human MHC. Was there any analysis for anti-human antibodies particularly human MHC? I am assuming that a single pool of human semen was utilized, is that true? This is because human semen contains variable levels of pro- and anti-inflammatory mediators (RAMETSE CL, VIRAL IMMUNOL 27: 200, 2014). Why were not the studies performed with rhesus seminal fluid? Please see VandeVoort C, Reprod Biol Endocrinol. 2: 33, 2004).
2. If indeed one is trying to mimic the human transmission studies, it would seem that a more fair comparison would be to use semen from HIV+/SIV+ donors. This becomes an important issue since there is clearly anti-HIV Ig in such seminal fluid that should be considered as a factor that could influence susceptibility in transmission studies (Pillay T, Front. Immunol. 9: 31`41, 2019).
3. There is also an important role for exosomes in seminal fluid that needs to be noted (Quattara LA, Andrologia Dec 2018).
4. Were there differences in susceptibility between groups 1 and 2? This was not clear in the text.
5. Line 78 is confusing, was the challenge with SIVmac251 contain semen? It states challenge with or without semen?
6. Lines 86-92, I do not understand why BJ39 was not excluded because we do not really know if SIV specific T cell immunity dies or does not contribute to resistance from infection susceptibility.
7. What happened to the animals remaining in each of the groups since only 8/12 or 8/11 were included in the challenge study? What was the criteria for exclusion/inclusion for the challenge study.
8. What is the basis for stating FoxP3 expression by CD4+ Tcells and therefore Tregs? Several lineages express FoxP3 including myeloid cells and even epithelial cells in select studies. Were co-localization studies performed? Not clear in the text.
9. It is stated that the animals in the 4 groups had similar MHC types. There appear to be 5/16 that had some of the naturally protective alleles in the Se+ group and 3/16 in the Se- group. Were the data analyzed to examine this issue?

Point-by-Point Response to Revision of NCOMMS-19-00438

We thank the reviewers for their thoughtful comments: below, are itemized answers to all the comments.

Reviewer #1

Major Points

- 1) *“Thus, they should compare the acquisition rate between semen /CEM (gp2) vs vehicle/CEM (gp4)...In the Fig. S4 A and B; They also compare Control /CEM group with Control /smB7 group. Is this control group similar to vehicle group? This is so confusing and need a clarification. Also, they should clearly explain why preconditioning needed for this study.”*

ANSWER:

- a. Note the original study analysis plan was designed as a 2x2 factorial grouping where each target condition (semen or smB7) was analyzed in groups of 16 as shown in Fig 1, whereas smaller groups of 8 animals each where treatments could be isolated were analyzed under secondary analysis (shown in supplementary data figure 4). The reviewer’s comment refers to the secondary analysis of isolated groups. We have now clarified the manuscript language to clarify data in Figure S4.
 - b. Regarding the explanation for preconditioning: as now stated in manuscript, we had two goals for this study. (1) To model the effects of continued semen exposure (with or without viral antigen stimulation) by using the macaque model. (2) To test the effects of preconditioning on susceptibility to viral infection (SIV). We have clarified these points in manuscript.
- 2) *“The effect of semen in the peripheral blood CD4 T cell is very significant finding in this study. However, In the Fig 2.B each group looks like mixture of a semen treated and untreated group. Can you have separated smB7 exposed animals from non-expose animals in each group? It is important to see the significant difference is not a result of exposure to smB7”*

ANSWER:

We now present a new Fig 2B where each subgroup is clearly distinguished. Data shows that levels of CCR5 in semen exposed groups are lower and that significant difference is not the result of exposure to SIVsmB7.

- 3) *“Didn’t you study the reduction of Ki67+CD4 T cells in the blood in the animals expose to semen?”*

ANSWER:

We did not study levels of Ki67 in circulating CD4 T cells but instead used Ki67 to stain tissues. We addressed peripheral blood activation of CD4 T cells by staining cells with CD25 and CD69 as common markers of activation.

- 4) *“Haven’t you study the Immune cell responses in lymph nodes (draining lymph nodes of FRT) of sacrificed animals?”*

ANSWER:

As we document an overall lack of adaptive T or B cell immune responses up to time-point before infection in all but 2 out of 32 animals, no lymph nodes were collected in sacrificed animals after they became viremic. Having established a lack of anti-SIV responses in animals up to the time before infection also lowers the expectation for any detectable immune cell responses in lymph nodes in animals exposed to preconditioning alone in absence of SIV challenge.

- 5) *“Immunohistochemistry data very convincing and shows that the effect of semen in tissue CD4 T cells and MX1 and Ki67 expression. However, sample numbers used in these graphs are not consistence. Fig 3 B, C, D: 6 animals in each group. In the Fig. 3 E, F, G: no-serum group has 6 and semen group has 8. Why this inconsistency. very confusing?”*

ANSWER:

Data shown represents all data that was available. Immunohistochemistry shown on Figure 3 represents all data from intact tissue blocks. Note staining data variance, when present, was due to censored tissue blocks due to poor quality sections. For example, one animal from Group 5 and one animal from Group 6 did not have usable endocervical tissue blocks. We have added a sentence in methods to confirm that all available data was reported.

- 6) *“The results of this tissues will be enhanced if they have stained for CXCR3 or IP10 to demonstrate the inflammatory status of the tissues. Have you stained for NK or macrophages, DC markers in the vaginal tissues? Authors discuss tissue associated immune mechanisms such as ADCC, so it is important to study tissue specific immune cells in addition to T cells (by flow cytometry or immune histochemistry). “*

ANSWER:

Unfortunately, our attempts to stain for CXCR3 or IP10 was not fruitful. We did stain FRT tissues for CD68+ macrophages as now displayed in supplemental figures 5 and 6, and CD123 as displayed in supplemental figure 5. Further analysis for changes in cell subset distribution is shown supplemental table 4. As our data was not collected to address repeated semen exposure effects on ADCC, we have removed discussion text on this point.

Minor Points

- 1) *“Fig. 1 D and E. It is important to see how each animal viral load and CD4 counts changes during the infection in each group. Thus, please present data in line charts.”*

ANSWER:

As requested by reviewer, we now present each animal viral load and CD4 count data as line charts in new Fig 1 panels E and F.

- 2) *“Page 5: line 40 “Taken together, our data demonstrate that repeated semen exposure results in higher FRT tissue MX1 expression, lower levels of CCR5 on CD4+ T-cells” have you shown CCR5 level on tissue specific T cells?”*

ANSWER:

Based on reviewer comment, we have modified this statement to focus on circulating CD4+ T-cells which more accurately reflects our data: *“Taken together, our data demonstrate that repeated semen exposure results in lower levels of CCR5 on circulating CD4+ T-cells...”*

- 3) *“Although not studied it is important to mention about the possible contribution of vaginal microbiome for the FRT immunological responses during semen exposure.”*

ANSWER:

We now include mention of the microbiome in discussion section with regards to important variables our data does not address.

Reviewer #2

Major Points

- 1) *“First is the question of whether the effects observed are semen specific, or alternatively are the effect of human semen in a macaque vagina. In other words, whether cross species immune responses are responsible for the altered immune responses and impact on SIV transmission. One previous finding that makes this concern particularly relevant is the vaccine studies from the early 1990s that demonstrated that a vaccine being produced in human cells was protective against a challenge SIV virus produced in human cells. However, it turned out that the protective immune responses were actually to human proteins, not SIV proteins.”*

ANSWER:

We thank reviewer for this important comment. Studies in early 1990s showed that vaccines produced in human cells generated immune responses against human MHC-II which conveyed protection against SIV challenge stocks also grown in human cells (virions carried human MHC-II). These results are not relevant to our study as our challenge stock was grown in rhesus PBMC absent any human antigen. We now include text to clearly note this point in the methods section as well as in main manuscript text. Furthermore, the earlier studies in early 1990s established strong immunogenicity following repeated IM vaccinations whereas in our study exogenous atraumatic exposure to semen within the vaginal lumen was shown to be poorly immunogenic as none of the animals exposed to semen and smB7 over 20 weeks developed anti-SIV responses. We also note in text (also listed in Supplementary Table S1) that CEM control supernatants inclusive of human MHC-I/II antigens were present in all pre-conditioning groups or all 32 animals challenged in study.

- 2) *"If there are immune responses being generated to human proteins or sperm then these may be indicative of a local immune environment in the vagina that is being modulated by human proteins."*

ANSWER:

We did not evidence any increase in immune activation in the vagina of semen exposed animals following assessment for changes in inflammatory cytokines and chemokines as collected by Weck-Cel sponges. We did not see any difference in multiple cytokines between semen and vehicle treated animals. No evidence of modulation of inflammation as a consequence human proteins was found. As noted in manuscript, we do interpret that the FRT does respond to human proteins in semen such as TGF-beta and prostaglandins would modulate a tolerogenic state of lower immune activation and macaque T-reg modulation.

- 3) *"Number 1, the authors hypothesize that TGF-beta and prostaglandins in semen might influence levels of local T-regs in the vagina(line 183). Is it known if human versions of these proteins can influence macaque T-regs?"*

ANSWER:

Recombinant human TGF-beta can modulate macaque T-regs as shown in the in vitro expansion of macaque CD4+ T regulatory cells [Doms et al. Am. J. Transplant, 2013. 13(8)]. This reference is now included.

- 4) *"Number 2, if there are immune responses being generated to human proteins or sperm then these may be indicative of a local immune environment in the vagina that is being modulated by human proteins."*

ANSWER:

As noted above, the exposure of semen was not observed to be immunogenic and the challenge virus used in our study was devoid of any human antigens.

- 5) *"Second, the manuscript is difficult to follow with regard to which group is getting which of the conditioning in their vaginas. Therefore, the suggestion is to better specify the different groups with some type of nomenclature that one can easily follow. For example, groups A, B, C, D could be used to designate the macaque groups that were SIV challenged.... In addition, a group designation of E, F, G, H could be utilized for the macaque groups that were not SIV challenged."*

ANSWER:

As recommended by reviewer, we now identify our study groups as Group 1 through 4 and distinguish the equally pre-conditioned but not challenged animals as groups 1a through 4a, respectively. This new group designation is described in ms. text and methods text.

- 6) *"Then for example, Figure 1 instead of trying to riddle which groups were included in the analysis or 'vehicle' or 'semen' it could say specifically which groups were evaluated."*

ANSWER:

To further aid readers, we have added a panel B in Figure 1 to illustrate the week of infection for each animal in each group.

- 7) *"Third, while assessing peripheral CCR5 CD4 cells is interesting, it would be even more interesting to quantifying the levels of SIV/HIV target cells in the cervix or vaginal area where the semen is being placed."*

ANSWER:

We did not have enough material to quantify the SIV target cells in the cervix where semen was placed as we prioritized sampling of soluble proteins by using weck cel sponges prior to semen dosing.

- 8) *" Fourth, the discussion and data addressing the potential mechanisms by which the semen might be achieving the SIV transmission reduction could be discussed in more detail (with less text designated for discussion of how the findings impact HIV studies)."*

ANSWER:

We have included a more detailed discussion of the potential mechanisms by which the semen might be achieving the SIV transmission reduction. In addition, we have reduced and limited extrapolation to how findings impact HIV studies to last paragraph in discussion.

Minor Points

- 1) *" Table S3 and figure 2 describe opposite findings. Figure 2 has the CCR5 lower in the semen treated macaques and table S3 has the CCR5 lower in the semen-naïve."*

ANSWER:

This unintended error has been corrected.

- 2) *"There have been observations that sex workers that take a break have an increased rate of HIV seroconversion soon after they return to work. I think these observations are important for the reasoning and understanding of why your experiments are relevant.... These findings should be described."*

ANSWER:

We agree this is an extremely relevant point, and we have added this to discussion.

- 3) *" Please define the term "high-level semen"*

ANSWER:

This statement has been clarified in text and defined as "five or more unprotected coital events in a week".

- 4) *" While the study provides clear evidence of FRT immune environment following semen exposure, it is over-speculation to suggest here that semen exposure could impact ADCC or vaccine protection outcome. Suggest limiting discussion to logical extrapolation of findings described."*

ANSWER:

We have removed all speculative language in the ADCC discussion, and we now limited the extrapolation of our data to HIV to the last paragraph in discussion.

- 5) *"While the data is generally presented with clarity (except as discussed above) the quality of the writing in manuscript is lacking. Spending some time refining the narrative would benefit the readers."*

ANSWER:

We have re-examined our narrative for additional edits that may benefit readers.

Reviewer #3

- 1) *"Overall, the protocol was difficult to follow and critically evaluate and details for parts of the study were missing. For sake of simplicity, I would suggest a simple change in nomenclature. The animals administered SIVsmB7 could be abbreviated to antigen (Ag) and those administered semen abbreviated to (Se). Thus, there would then be 4 groups of animals, group 1 Se+/Ag+, group 2 Se+/Ag-, group 3 Se-/Ag+ and group 4 Se-/Ag-- throughout the text OR something along these lines."*

ANSWER:

As noted above, we have clarified protocol by introducing a new classification for Groups as 1-4. Figures and text have been modified to reflect this change.

- 2) *"It seems that would clearly be some sort of xenogeneic response to human proteins including human MHC. Was there any analysis for anti-human antibodies particularly human MHC? I am assuming that a single pool of human semen was utilized, is that true? This is because human semen contains variable levels of pro- and anti-inflammatory mediators (RAMETSE CL, VIRAL IMMUNOL 27: 200, 2014). Why were not the studies performed with rhesus seminal fluid? Please see VandeVoort C, Reprod Biol Endocrinol. 2: 33, 2004."*

ANSWERS:

- a. As noted above in response to reviewer 2, we address this concern by (1) clarifying the source of challenge stock of virus used was grown in primary rhesus PBMC absent any human MHC antigens, (2) adding text on the lack of inflammatory or anti-inflammatory responses documented pre-challenge in 20 week semen exposed cervicovaginal environments, and (3) reviewing the overall lack of anti-SIV antibody responses with semen in SIV B7 exposed animals to document an overall lack of antigenicity during preconditioning periods.
 - b. To limit variability in human semen samples, we prepared and used a single pool at start (containing dozens of unique samples) for the entire duration of the study. This is now clearly stated in methods.
 - c. Our initial attempts to collect a suitable pool of macaque semen were abandoned due to: (a) adult male animal resistance to rectal probe electrostimulation to induce ejaculate, and (b) when successful in inducing ejaculate, a very limited/variable yield of seminal fluid per ejaculate was available as macaque semen forms a solid "plug" reducing the liquid portion.
- 3) *"If indeed one is trying to mimic the human transmission studies, it would seem that a more fair comparison would be to use semen from HIV+/SIV+ donors. This*

becomes an important issue since there is clearly anti-HIV Ig in such seminal fluid that should be considered as a factor that could influence susceptibility in transmission studies (Pillay T, Front. Immunol. 9: 31`41, 2019)."

ANSWER:

We completely agree with the Reviewer's comment that factors such as anti-HIV Ig could contribute to transmission. Note that our stated objective was to assess the effects of repeated semen exposure on the FRT (with or without viral particles) as well as its impact on infection rather than model all aspects of transmission. Indeed, our data suggests that repeated exposure to semen from uninfected males could mediate the effect described. However, we now directly state that among the limitations of our work is the fact that we do not address effects of semen from HIV positive viremic/suppressed persons or the potential for cell-associated viral transmission.

- 4) *"There is also an important role for exosomes in seminal fluid that needs to be noted (Quattara LA, Andrologia Dec 2018)."*

ANSWER:

We have highlighted this important point in discussion.

- 5) *"Were there differences in susceptibility between groups 1 and 2? This was not clear in the text."*

ANSWER:

We did not find any significant difference between these two groups (log-rank $p = 0.24$). We have added Figure S4 panel C to show this comparison.

- 6) *"Line 78 is confusing, was the challenge with SIVmac251 contain semen? It states challenge with or without semen?"*

ANSWER:

We have edited text for clarity, stating that all animals conditioned with semen had their challenge virus also diluted in semen to maintain effects. Table S1 has also been edited to clarify this point.

- 7) *"Lines 86-92, I do not understand why BJ39 was not excluded because we do not really know if SIV specific T cell immunity does or does not contribute to resistance from infection susceptibility."*

ANSWER:

To our knowledge, no NHP study to date has demonstrated SIV protection or resistance via a CD8+ T cell responses alone. By contrast, various studies have shown resistance or protection from SIV/SHIV from passive administration or vaccine induced anti-SIV/SHIV antibody responses. In spite of these detective responses, the animal in question BJ39 was readily infected by week 4 of the study as highlighted in new Fig. 1B.

- 8) *"What happened to the animals remaining in each of the groups since only 8/12 or 8/11 were included in the challenge study? What was the criteria for exclusion/inclusion for the challenge study."*

ANSWER:

While we are not certain where reviewer derives the 8/11 or 8/12 reference, we confirm that all 32 animals from Groups 1 through 4 were included in the challenge study post-conditioning. As stated in text, 31 out of 32 animal infection outcomes were used in analysis. 11 animals not infected at the lower dose challenge proceeded to a high dose challenge to confirm infectivity.

- 9) *“What is the basis for stating FoxP3 expression by CD4+ T cells and therefore T regs? Several lineages express FoxP3 including myeloid cells and even epithelial cells in select studies. Were co-localization studies performed? Not clear in the text.”*

ANSWER:

Based on the Reviewer request, we now provide new data demonstrating that FoxP3 staining is isolated to CD3+ T cells (not myeloid cells) in supplementary Figure 8. Foxp3 positive cells were localized in the lamina propria and not in the epithelial layer.

- 10) *“It is stated that the animals in the 4 groups had similar MHC types. There appear to be 5/16 that had some of the naturally protective alleles in the Se+ group and 3/16 in the Se- group. Were the data analyzed to examine this issue?”*

ANSWER:

To our knowledge there is no known MHC allele associated with protection from infection in the rhesus macaque model. Alleles have been associated with spontaneous control of viremia, yet our study goal was designed to examine susceptibility to infection. Therefore, we did not specifically assess infection differences based on allele distribution. Of interest, the 3/16 animals noted by reviewer in Se- group were all infected during the low-dose challenge period.

Reviewers' Comments:

Reviewer #1:

Remarks to the Author:

The authors have addressed all the issues raised

Reviewer #2:

Remarks to the Author:

My comments have now been addressed. Very interesting and potentially important study.

Reviewer #3:

Remarks to the Author:

I believe that the authors have addressed most of the issues in a satisfactory manner. However, one issue still remains. This reviewer has trouble with the use of xenogeneic proteins. The authors rightfully state that the same proteins were also administered to the control monkeys and that the fact that there were no anti-SIV antibody responses together suggest the lack of "immunogenicity" of the xenogeneic proteins. I think it would be prudent for the authors to first of all state in the discussion whether immune responses against the xenogeneic semen proteins was or was not measured and that it is quite possible that the response to SIV could be modulated by the exposure to xenogeneic proteins contributing to the results observed. In other words, response does not necessarily mean antibody responses, they include tolerogenic responses for instance. A short sentence addressing this issue in the discussion should suffice.

Reviewer Point-by-Point Response to Revision of NCOMMS-19-00438A

We thank the reviewers for their thoughtful comments: below, are itemized answers to all the comments.

Reviewer #1

"The authors have addressed all the issues raised."

ANSWER:

We thank reviewer for helping us improve the manuscript.

Reviewer #2

"My comments have now been addressed. Very interesting and potentially important study."

ANSWER:

We thank reviewer for helping us improve the manuscript.

Reviewer #3

"I believe that the authors have addressed most of the issues in a satisfactory manner. However, one issue still remains. This reviewer has trouble with the use of xenogeneic proteins. The authors rightfully state that the same proteins were also administered to the control monkeys and that the fact that there were no anti-SIV antibody responses together suggest the lack of "immunogenicity" of the xenogeneic proteins. I think it would be prudent for the authors to first of all state in the discussion whether immune responses against the xenogeneic semen proteins was or was not measured and that it is quite possible that the response to SIV could be modulated by the exposure to xenogeneic proteins contributing to the results observed. In other words, response does not necessarily mean antibody responses, they include tolerogenic responses for instance. A short sentence addressing this issue in the discussion should suffice."

ANSWER:

As requested by reviewer, we have added a sentence to paragraph stating limitations of work to highlight that immune responses against xenogeneic semen proteins was not measured.